# From QKV to K/KV: Investigating Minimalist Attention Mechanisms

## Abstract

Transformers have become the standard solution for various AI tasks. The widely adopted query, key, and value (QKV) formulation has played a significant role in this. Although the performance of transformer models has been widely studied, the individual contribution of these three components and the precise impact on performance when some are omitted are still not fully understood. Consequently, we evaluated two transformer variants: one with two projections to construct K and V vectors, and another with only a single projection. Both resulted in symmetric self-attention maps. Additionally, we explored an asymmetric attention mechanism by incorporating a 2D positional encoding into the attention matrix. In particular, these modified transformers exhibited reduced parameter counts and computational demands compared to the standard architecture. Through experiments encompassing three task types: synthetics (such as reversing or sorting a list), vision (MNIST, CIFAR, and Tiny ImageNet classification) and NLP (character generation and translation)—we discovered that our transformers perform on par or occasionally better than the QKV transformer on vision tasks but under-perform slightly on NLP tasks. Our findings suggest that three distinct self-attention representations are not universally required and depend on the specific task.

## 1    Introduction

Transformers (Vaswani et al., 2017) have gained significant attention in recent years due to their effectiveness in various domains such as language, vision, and audio processing (Han et al., 2022). The research community has witnessed a notable increase in the number of variants of the Transformer model. These "X-former" models, such as Reformer, Linformer, Performer, and Longformer, as well as Ring attention, Blockwise attention, and sparse attention mechanisms, aim to enhance the original Transformer architecture with advances in computational and memory efficiency (Tay et al., 2022).

Although most Transformer architectures rely on query, key, and value, the necessity of all three components remains unclear. There is an apparent redundancy of representations in Transformers compared to the more singular representations in RNNs and CNNs. To explore this, we propose and evaluate two simplified Transformers: KV (using key-value vectors) and K (using only key vectors). Our results show that using fewer vectors reduces the complexity of the model, the number of parameters, and the computational cost while largely maintaining the performance across various tasks. The extent of any performance decrease is task-specific; for instance, image classification, lacking a temporal dimension, benefits from symmetric attention (since K and Q are the same). Notably, even in sequential and temporal tasks, our reduced-projection approach achieves strong performance with fewer parameters and less computation.

Our research paves the way for more efficient Transformer designs, essential for resource-constrained devices, and offers key insights into the mechanics and interpretability of self-attention.

## 2    Related Work

To address the computational demands of transformer models, researchers have extensively explored techniques like pruning, quantization, sparsification, and weight sharing. **Pruning** aims to reduce the model size and inference cost by removing redundant connections or parameters. For instance,

SparseGPT (Frantar & Alistarh, 2023) demonstrates one-shot pruning of large GPT models with minimal accuracy loss. Movement pruning (Sanh et al., 2020) and block pruning (Lagunas et al., 2021) offer structured approaches to enhance hardware efficiency. **Quantization** lowers the precision of weights and activations, leading to smaller models and faster computations. Post-training quantization (PTQ) and quantization-aware training (QAT) are common strategies (Zadeh et al., 2020). Techniques like SmoothQuant (Xiao et al., 2023) and OmniQuant (Wu et al., 2023) address challenges related to outlier activations. **Sparsification** introduces sparsity into weight matrices, often through magnitude-based pruning (Zhu & Gupta, 2017). SparseML (Magic, 2023) is a toolkit that facilitates the application of sparsification recipes. The combination of these techniques, such as joint pruning and quantization (Shen et al., 2020; Intel, 2023), presents further opportunities for efficient transformer deployment. Recent surveys (Yao et al., 2024; He & Liu, 2024) provide comprehensive overviews of these transformer compression methods.

Research has consistently demonstrated the value of **weight sharing** in optimizing Transformer models for improved efficiency and reduced complexity. Early investigations, such as "Sharing Attention Weights for Fast Transformer" (Xiao et al., 2019), focused on accelerating inference by sharing attention weights across layers. Subsequent research has explored more sophisticated weight sharing strategies to minimize potential performance degradation. For instance, "Residualtransformer" (Wang & Li, 2024) shares a full-rank component while retaining unique low-rank components in each layer, and "MiniViT" (Zhang et al., 2022) introduces weight multiplexing to enhance the diversity of shared weights, particularly in Vision Transformers (Dosovitskiy et al., 2021). "Subformer" (Reid et al., 2021) combines sandwich-style sharing with self-attentive factorized embeddings to improve parameter efficiency in generative Transformers. Beyond these specific approaches, several studies have broadly examined weight sharing in Transformer architectures. ALBERT achieved significant model size reduction by sharing parameters across all Transformer layers and embedding matrices while maintaining competitive performance (Lan et al., 2019). The Universal Transformer recurrently applies the same Transformer block, effectively sharing weights across time steps and enabling adaptive computation (Dehghani et al., 2018). In multilingual models, Blackwood *et al.* explored sharing attention and feed-forward layers across language pairs to enhance translation quality with fewer parameters (Blackwood et al., 2018). Reformer also incorporated weight sharing alongside other memory-efficient techniques to scale Transformers to longer sequences (Kitaev et al., 2020), and TinyBERT combined weight sharing with knowledge distillation to compress BERT models for resource-constrained environments (Kitaev et al., 2020). These efforts illustrate that while naive weight sharing can be detrimental, carefully designed sharing schemes offer a promising avenue for reducing the computational demands of Transformers.

In addition to the parameter-sharing approaches, alternative designs such as multi-query and grouped-query attention (Shazeer, 2019; Ainslie et al., 2023) reduce the number of key/value projections across heads, leading to lower memory usage and faster decoding. More recently, "decoupled-head sharing (DHA)" (Chen et al., 2024) provides a flexible mechanism for parameter reduction and efficiency without overly constraining representational capacity. These strategies are highly relevant alternatives to weight sharing, as they also aim to reduce parameters while improving inference efficiency.

Finally, several works have focused on "positional-bias methods" that induce asymmetric attention scores with negligible parameter overhead. Learned relative position bias (Shaw et al., 2018), T5's relative bias (Raffel et al., 2020), ALiBi (Press et al., 2021), rotary position embeddings (RoPE) (Su et al., 2021), and two-dimensional relative biases in vision backbones (Liu et al., 2021) are widely used baselines for capturing relative positional structure. These methods provide important context when evaluating novel score-augmentation mechanisms that aim to introduce asymmetry into attention computations.

## 3    SELF ATTENTION WITH REDUCED PROJECTIONS

Transformer blocks are known for incorporating several key components, including a multi-head self-attention mechanism, a position-wise feed-forward network, layer normalization modules, residual connections, and positional encoding.

The self-attention mechanism, also known as intra-attention, is a crucial and distinguishing feature of Transformer models. Its purpose is to establish relationships between different positions within a sequence, enabling the computation of a representation for that very sequence. This mechanism has

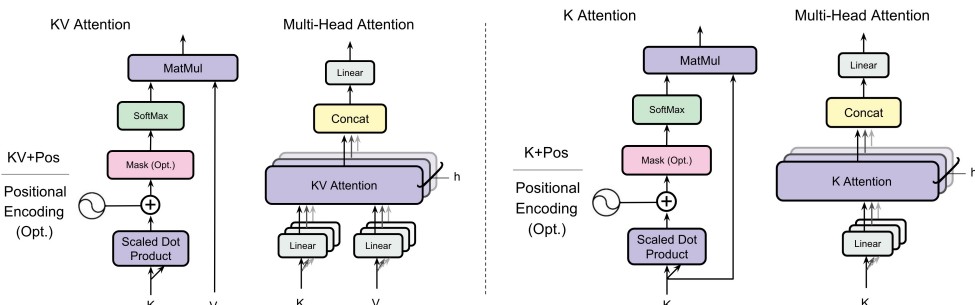

Figure 1: Left: KV Transformer using only K and V vector representations, Right: K Transformer (using only one vector representation). Attention with 2D positional encoding is denoted as X+Pos.

proven to be highly valuable in various tasks such as machine translation, abstractive summarization, and image description generation.

The fundamental concept underlying self attention mechanism is for each element within the sequence to learn the ability to gather information from other tokens present in the same sequence. The operation for a single head can be defined as follows:

$$A_h = \text{Softmax}(\alpha Q_h K_h^T)V_h, \tag{1}$$

where X is a matrix in $\mathbb{R}^{n \times d}$, $\alpha$ is a scaling factor typically set to $1/\sqrt{d_k}$, $Q_h = XW_q$, $K_h = XW_k$, and $V_h = XW_v$ are linear transformations applied on the temporal dimension of the input sequence. $W_q, W_k, W_v \in \mathbb{R}^{d \times d/H}$ are the weight matrices (parameters) responsible for the query, key, and value projections, respectively. These matrices project the input X into an output tensor of $d_k = d/H$ dimensions. $H$ represents the number of heads, which is assumed to be equal to 1 for the rest of the analysis. Softmax is applied row-wise.

The heads $A_1 \cdots A_H$ are computed in parallel, and their outputs are concatenated and passed through a dense layer. The attention matrix $QK^T$ is primarily responsible for learning the alignment scores between the tokens in the sequence. Notably, this formulation involves taking the dot product between each query (Q) vector and every key (K) vector.

### 3.1 PROPOSED TRANSFORMERS

We examine two variations of QKV self-attention. The first ties together the mechanisms for query and key, while the second unifies the processing of query, key, and value. These two approaches lead to the Transformer types we describe next.

- **KV Transformer:** In our formulation (Figure 1; left), we simply replace Q with K (*i.e.* Q is dropped), resulting in:

$$A = \text{Softmax}(\alpha KK^T)V. \tag{2}$$

  This attention mechanism features a symmetric 2D matrix. To address inherent limitations of symmetry, we introduce a "KV+Pos" variant as a comparative baseline. To introduce asymmetry, a 2D positional encoding of dimension $m$ (*i.e.* a tensor of size $n \times n \times m$; sinusoidal basis) is added to the $n \times n$ attention matrix. The resulting tensor, with dimensions $n \times n \times m$, is then projected back into a $n \times n$ matrix using a linear layer consisting of $m$ parameters.

- **K Transformer:** Here (Figure 1; right), only one projection is used: $A = \text{Softmax}(\alpha KK^T)K$. In a similar vein to the KV Transformer, we also examine an asymmetric attention variant known as the "K+Pos" Transformer.

Table 1 demonstrates the computational complexity of the two attention mechanisms versus the QKV attention. For the KV (symmetric) attention, computational complexity of linear projections of Q and K vectors (excluding the quadratic computation in $KK^T$) and the number of parameters (ignoring biases) is two-thirds that of the QKV attention. The number of parameters in KV+Pos attention is still lower than QKV attention ($m << d^2$). However, the drawback of the KV+Pos attention is that its computational complexity depends on $n^2$. Its computational cost is lower than QKV attention

Table 1: Comparison of the proposed Transformers and QKV Transformer in terms of computational complexity and number of parameters. $d$ is the embedding dimensionality, $n$ is the sequence length, and $m$ is the dimensionality of the 2D positional encoding layer. Note that computational complexity only includes the linear projections, and not the dot product computation in the self attention matrix. 2D positional embedding is not learned.

| | Computational Complexity | # Parameters |
|---|---|---|
| QKV | $3nd^2$ | $3d^2$ |
| KV | $2nd^2$ | $2d^2$ |
| KV+Pos | $2nd^2 + n^2m$ | $2d^2 + m$ |
| K | $nd^2$ | $d^2$ |
| K+Pos | $nd^2 + n^2m$ | $d^2 + m$ |

when $nm < d^2$. For instance, with $m = 100$ and $d = 1000$, KV+Pos attention is more efficient for sequence lengths below $10,000$. The K Transformer requires only one-third of the parameters and computation of projections in the standard QKV attention.

The choice of $m$, whether the addition of positional encoding is beneficial and whether symmetric KV attention is sufficient, all depend on the specific problem being addressed. Notice that our formulation provides a trade-off between model complexity and model performance, which becomes particularly important during inference time.

## 3.2 REMARKS

Multiplying K by its transpose leads to large diagonal activations in the attention map, possibly reducing the influence of off-diagonal relationships. We attempted to mitigate this using normalization techniques, such as dividing diagonal elements by their sum, but this did not yield consistent or substantial improvements. The appendix contains the corresponding results.

It is important to mention that certain tasks, such as translation, may necessitate the use of cross-attention. In such cases, we retain the QKV attention mechanism when needed, but replace self-attention with KV (or K) attention. Self-attention refers to the scenario in which the keys and values are derived from the same source as the queries. On the other hand, in cross-attention, the queries are still generated from the input sequence, but the keys and values are obtained from an external source, such as an encoder module.

Although our Transformers are more parameter and computationally efficient, the improvements are modest because self-attention projection parameters, the area of our focus, only represent about a third of a typical Transformer's total parameter count. However, their simplified designs contribute to a better understanding of self-attention mechanisms and further optimizations. For example, the inherent symmetry within the attention matrix lends itself to hardware optimizations or batch processing to enhance efficiency. In addition, previous research has shown that the softmax function might be dispensable in self-attention (Lu et al., 2021; Koohpayegani & Pirsiavash, 2024). Eliminating it could further simplify our attention mechanism, potentially yielding even greater efficiency. We defer the investigation of these possibilities to future works. Our approach is orthogonal to existing methods and can therefore be used in tandem with them.

## 4 EXPERIMENTS AND RESULTS

We conducted empirical experiments on three types of tasks including a) Synthetic (5 tasks), b) Vision (6 tasks), and c) NLP (3 tasks). All models are trained from scratch, except for the set anomaly detection. Our objective is not to achieve state-of-the-art performance, but rather to compare the attention mechanisms employed. The experiments were conducted on a workstation featuring an NVIDIA GeForce GTX 1080 Ti, a GPU with 11 GB of memory.

Some of the tasks are sequence-to-sequence, where the input and the output is a sequence, not necessarily of the same length. An example of tasks in this domain is translation. Here, a Transformer encoder is used for interpreting the input sequence and a decoder is used for generating the output in an autoregressive manner. Some other tasks include image classification and anomaly detection. In these tasks, only an encoder is used to map the input to a set of labels.

| | Reverse | Sort | Sub | Swap | Copy | Avg. |
|---|---|---|---|---|---|---|
| QKV | 0.698 | 0.971 | 1.0 | 0.588 | 1.0 | 0.851 |
| KV | 0.705 | 0.967 | 1.0 | 0.597 | 1.0 | 0.854 |
| KV+Pos | 0.718 | 0.963 | 1.0 | 0.671 | 1.0 | **0.870** |
| K | 0.514 | 0.939 | 1.0 | 0.446 | 1.0 | 0.780 |
| K+Pos | 0.581 | 0.957 | 1.0 | 0.576 | 1.0 | 0.823 |

Table 2: The performance of transformers on synthetic tasks. Multiple runs, over different configurations (such as number of attention heads, embedding dimension, learning rate, sequence length, etc.), are conducted, and the results are averaged.

## 4.1 SYNTHETIC TASKS

We focus on five specific tasks outlined below. The input list, which has a predetermined length, consists of numbers ranging from 0 to 9, inclusive of both 0 and 9.

**Reverse.** In this task, a list of numbers is subjected to a reversal operation. For instance, the input list [4, 3, 9, 8, 1] would be transformed into [1, 8, 9, 3, 4].

**Sort.** The objective of this task is to arrange the input list in ascending order. For example, [4, 3, 9, 8, 1] would be transformed into [1, 3, 4, 8, 9].

**Sub.** In this case, each element of the list is subtracted from 9. For example, the array [4, 3, 9, 8, 1] would be transformed into [5, 6, 0, 1, 8].

**Swap.** In this scenario, the first half of an even-length list is exchanged with the second half. For instance, the list [4, 3, 9, 8, 1, 7] would be transformed into [8, 1, 7, 4, 3, 9].

**Copy.** In this task, the objective is to retain the input list as is. For example, [4, 3, 9, 8, 1] remains unchanged as [4, 3, 9, 8, 1].

Here, only one transformer encoder is used. In training, we feed the input sequence into the encoder to generate predictions for each token in the input. We utilize the standard cross entropy loss for this purpose. Each number is encoded as a one-hot vector. We apply a gradient clip value of 5 and set the 2D positional embedding dimension to 10 (*i.e. m*). Additionally, we employ the Adam optimizer along with the CosineWarmupScheduler, using a warm-up period of 5.

We perform experiments with different configurations of transformer models by varying the embedding dimension (32, 64, 256), the number of layers (2, 4), the number of heads (2, 4), a learning rate of 1e-3 and the input sequence length (16, 64, 128). Each configuration is run three times for two epochs, and the results are then averaged across the configurations.

The QKV transformer exhibits faster convergence compared to the K and KV transformers (see loss curves in the Appendix). However, all transformers demonstrate good performance on synthetic tasks, as indicated by the accuracies presented in Table 2. The KV transformer achieves performance comparable to that of the QKV transformer, whereas the K transformer performs considerably worse. Incorporating positional information (X + Pos) substantially boosts the performance of our X transformer types.

Sample self attention maps over synthetics tasks are shown in the Appendix.

## 4.2 VISION TASKS

We evaluated performance on various vision tasks, including image classification in MNIST (LeCun et al., 1998), FashionMNIST (Xiao et al., 2017), CIFAR-10 (Krizhevsky et al., 2009), CIFAR-100 (Krizhevsky et al., 2009), and Tiny ImageNet (200 classes[1]), as well as anomaly detection.

**Classification**. We explore various settings for patch size (4, 7), learning rate (1e-3, 1e-4), embedding dimension (64, 256, 512), number of layers (2, 4), and number of heads (2, 4). For each configuration, we performed two experiments, each experiment lasting $k$ epochs. The value of $k$ differs depending on the dataset: 20 epochs for MNIST and FashionMNIST, 40 epochs for CIFAR-10, and 50 epochs

---

[1]https://paperswithcode.com/dataset/tiny-imagenet

|       | MNIST | F-MNIST | CIFAR-10 | CIFAR-100 | TinyImageNet | Anomaly | Avg.  |
|-------|-------|---------|----------|-----------|--------------|---------|-------|
| QKV   | 0.981 | 0.887   | 0.663    | 0.363     | 0.229        | 0.942   | 0.767 |
| KV    | 0.981 | 0.885   | 0.666    | 0.369     | 0.236        | 0.954   | 0.771 |
| KV+Pos| 0.982 | 0.884   | 0.662    | 0.366     | -            | 0.966   | **0.772** |
| K     | 0.978 | 0.877   | 0.672    | 0.376     | 0.266        | 0.933   | 0.767 |
| K+Pos | 0.977 | 0.875   | 0.669    | 0.364     | -            | 0.961   | 0.769 |

Table 3: The performance of transformers on vision tasks. The average column does not include TinyImageNet.

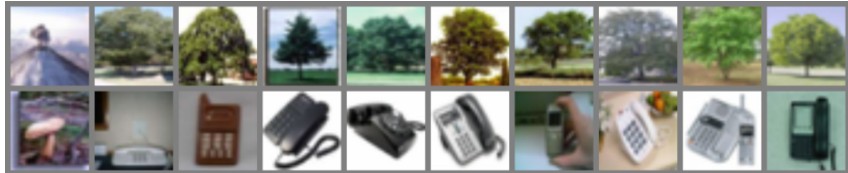

Figure 3: Two sets of samples from the anomaly detection dataset, with the first image in each set representing the anomaly.

for CIFAR-100. We employ the cross-entropy loss function and utilize the Adam optimizer with the MultiStepLR scheduler for optimization. In the case of 2D positional encoding, we set pos dim to 50.

As indicated in Table 3, the KV + Pos transformer exhibits performance comparable to that of the QKV transformer in the MNIST, FashionMNIST and CIFAR datasets. The K transformer, while slightly behind these two variants on MNIST and FashionMNIST, still performs at a reasonably competitive level on CIFAR datasets.

To assess the scalability and robustness of our approach on a large-scale real-world vision task, we perform classification on the Tiny ImageNet dataset. This dataset contains 100K images of 200 classes (500 for each class). Each class has 500 training images, 50 validation images, and 50 test images. We use a Vision Transformer (ViT) model that is configured with the following parameters: image size of 224, patch size of 16, 200 classes, embedding dimension of 768, 12 layers, 12 attention heads, MLP dimension of 3072, and a dropout rate of 0.1. The optimization process and loss function are as above.

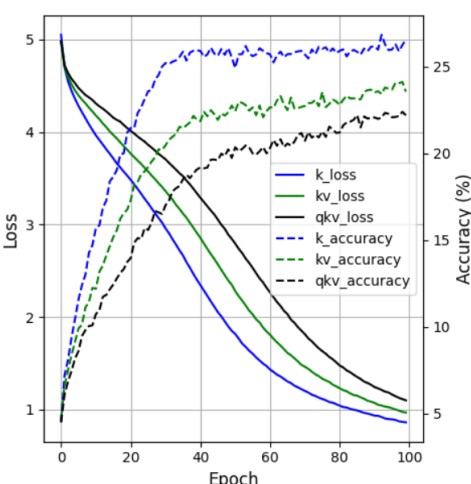

Figure 2: Training loss and validation accuracy for TinyImageNet image classification.

All models were trained from scratch (*i.e.* no use of pretrained backbones). We evaluate three self-attention variants—QKV, KV, and K—each run twice. Figure 2 shows the training loss and validation accuracy over epochs. Numerical results are provided in Table 3. The corresponding training times per epoch are 40, 35, and 32 minutes on GPU, demonstrating improved efficiency with small impact on accuracy. Notably, the K Transformer, despite employing only one projection, achieves strong results in this instance. Continued training over more epochs could potentially close the performance gap between the Transformer architectures.

**Set Anomaly Detection.** Here, we aim to apply transformers to sets (*i.e.* unordered inputs). A model is tested for its ability to find the odd one out in a set of ten images, using CIFAR-100. Nine images are from one class, and one is different. Two sample sets are shown in Figure 3. CIFAR-100 comprises 60K 32x32 images over 100 classes (600 per class).

To extract high-level, low-dimensional features from the images, we employ a pre-trained ResNet34 model (He et al., 2016) pretrained on the ImageNet dataset (Deng et al., 2009). To monitor the training progress and determine when to stop, a validation set is created. In this scenario, we divide the training set into 90% for training purposes and 10% for validation, ensuring a balanced distribution across classes.

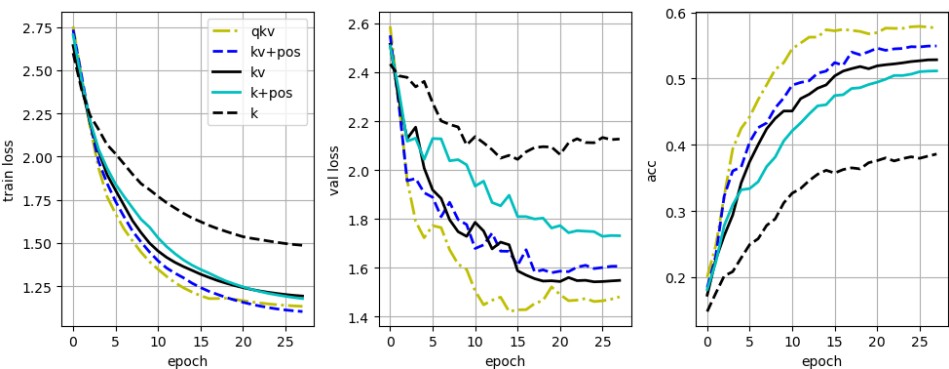

Figure 4: Epoch-wise loss and accuracy in the number generation task.

We define an epoch as a sequence in which each image within the dataset is considered as an "anomaly" exactly once. Therefore, the length of the dataset is determined by the total number of images it contains. When constructing the training set, we follow a two-step process. First, we randomly sample a class that is different from the class of the image at the corresponding index (*i.e.* __getitem__(self, idx)). Then, in the second step, we sample 9 images from the newly selected class.

We perform a set-level classification, outputting a single logit per image to ensure permutation equivariance in our predictions. A softmax function is applied across these image-specific logits, and the model is trained to assign the highest probability to the anomalous image. This differs from standard classification where softmax operates over class outputs. Consequently, if the order of input images is changed, their corresponding probabilities in the softmax output are also reordered, thus achieving permutation equivariance.

In our experiments, we vary the embedding dimension, selecting from the options of 256 and 512. Additionally, we explore different depths and numbers of heads, choosing values of 2 and 4. We set the learning rate to 5e-4 for all configurations. We incorporate a dropout rate of 0.1 throughout the model to facilitate regularization. To control the model's learning rate, we utilize the CosineWarmupScheduler. We configure the warm-up parameter (set to 100) to gradually initiate the model training process. Each setting is executed twice for a total of 20 epochs, and the results are subsequently averaged to obtain reliable performance measurements (see Table 3).

The last column of Table 3 presents the results of this experiment. It shows comparable performance across models, with KV+Pos exhibiting a slight advantage.

### 4.3 NLP TASKS

In this section, we explore three NLP tasks: number generation and character generation, which both use only an encoder, and language translation, which uses an encoder-decoder architecture.

**Number generation.** In this task, our objective is to generate the next token in a dataset that comprises written numbers. The dataset includes consecutive numbers from 1 to 9999, spelled out as words. The task involves predicting the subsequent token given the preceding $l$ tokens. For instance, if the sequence length is $l = 3$, we would have examples like these:

$$(['thirteen', '.', 'fourteen'], '.'), \quad (['.', 'fifteen', '.'], 'sixteen'), \quad \cdots$$

The vocabulary size for this dataset is 30, and there are a total of 63,095 tokens.

For transformer training, we explored sequence lengths ($l$) of 16, 32, and 64, and varied the number of heads between 2 and 4. We fixed the learning rate at 1e-2, the positional dimension at 10, and the embedding dimension at 64. Each parameter combination was trained twice for 30 epochs, with cross-entropy loss.

The results are depicted in Figure 4. We observe a performance decline using KV and K Transformers. Nevertheless, including positional information in the attention matrix does help retain some performance. Because the K and K+pos models yielded poor results here, we will discard them in the next set of experiments.

**Character generation.** The objective in this task is to generate text by predicting the next character. We used the tiny Shakespeare dataset, which consists of 1,115,394 characters. For the training process, we allocate 90% of the data for training purposes, while the remaining portion is used for validation. During each training step, we track the loss across the entire validation set.

Several hyperparameters are considered, including context size (8, 32, 64), maximum iteration set to 1000, learning rate of 5e-4, embedding dimension chosen from 64 and 192, number of heads selected from 1 and 4, number of layers chosen from 1, 2, 3, and 6, dropout rate set to 0.2, and positional dimension set to 20. The optimization process involves using the Adam optimizer along with cross entropy loss. In total, four models are trained. The average results are illustrated in Figure 5. KV+Pos and KV transformers show similar learning speeds, while QKV converges faster. However, all three ultimately reach a comparable level of val loss, suggesting a similar final performance. Qualitative assessment indicates that all models can generate coherent sentences.

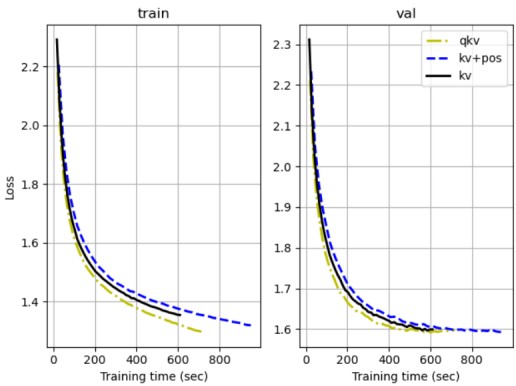

Figure 5: Loss in character generation task.

**Translation.** The objective here is to train a transformer model from scratch to translate sentences between two languages. Specifically, we used the Multi30k dataset (Elliott et al., 2016) to train a German to English translation model as well as an English to German translation model.

In contrast to using a one-hot target distribution, we adopt a different approach. We set the probability of the target word to a predetermined "confidence value" (typically 0.9) and allocate the remaining "smoothing value" mass (usually 0.1) evenly across the rest of the vocabulary. This technique, known as label smoothing, aims to provide a more robust training signal.

To optimize the model, we employ the KL divergence loss and the Adam optimizer. The learning rate follows a linear ramp-up for a specified number of warm-up steps (usually 4000) and then decays according to the inverse square root law based on the current training step number. During the translation process, we utilize a greedy decoding strategy, starting with a designated start token.

For the model configuration, we set the positional dimension to 10 and apply a dropout rate of 0.1. We explore different variations by varying the number of layers (1 or 2), the number of heads (1 or 4), and the embedding dimension (64, 128, or 256). Each variant is trained twice for 15 epochs. Loss values and BLEU scores (Papineni et al., 2002) are recorded.

The results for both German-to-English and English-to-German translations are depicted in Figure 6. Interestingly, incorporating 2D positional encoding negatively affects the model's performance. However, the KV transformer demonstrates competitive performance compared to the QKV transformer in this task. The performance gap could stem from our Transformers' use of cross-attention, disrupting the uniform self-attention used across the network.

## 5 DISCUSSION AND CONCLUSION

We conducted an evaluation comparing the performance of self attention with reduced projections, both with and without 2D positional encoding, to the commonly used QKV attention across 14 different tasks. Our overall results imply that the necessity of three separate representations in self-attention for an input sequence is task-dependent; it appears not to be essential for at least some tasks we examined. In certain cases, such as the synthetic and vision datasets, KV attention outperforms QKV attention. It is worth noting that there is a trade-off involved: KV and K attentions (without positional encoding) achieves lower accuracy but require fewer parameters compared to QKV attention.

Our primary objective was not to achieve state-of-the-art performance by fine-tuning or optimizing the models, but rather to investigate whether there is a notable difference in performance between the proposed transformer architectures and the original QKV (Query-Key-Value) Transformer. In

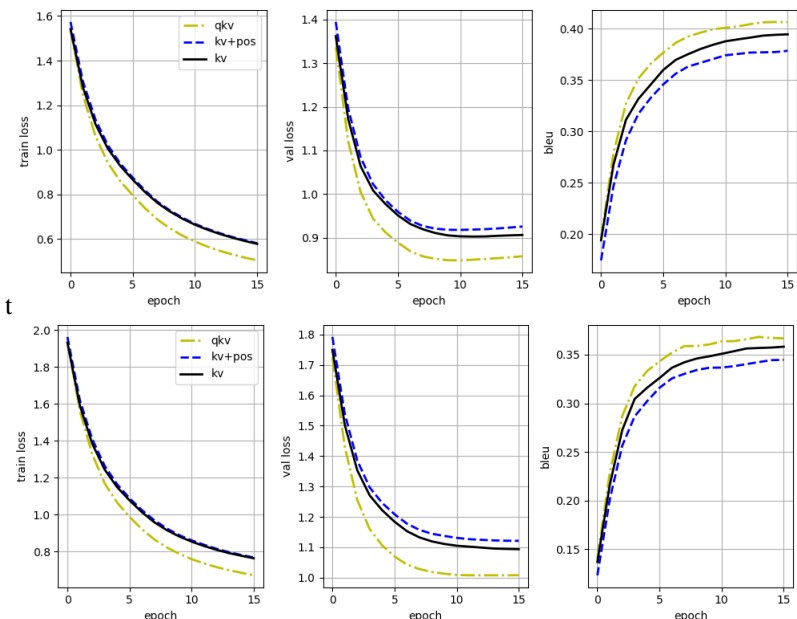

t

Figure 6: Loss and BLEU score per epoch in the translation task. BLEU is computed over the test set. Top: German to English, Bottom: English to German.

addition to this, our further analysis demonstrates that the insights gained from our study have broader applicability. Specifically, we found that our results are not limited to a single task but can extend to other vision-related challenges, such as semantic segmentation, showing that the proposed transformers may hold promise for improving performance in a variety of computer vision tasks.

Models show similar performance on synthetic tasks, possibly because these tasks are simpler and involve fewer tokens. Large-scale vision tasks also exhibit comparable model performance. However, in NLP tasks, dropping projections seems to lead to a performance decrease. The use of separate linear layers to generate Query, Key, and Value vectors enables the model to capture various aspects of each token and how they contextually relate. This is highly advantageous for sequential tasks where token order influences meaning and for temporal data. However, our results show that for tasks such as image classification, this level of attention complexity might not be required, implying that simpler and more resource-friendly Transformer designs are feasible depending on the application.

Future work could explore novel techniques for making Transformers more efficient and generalizable. Furthermore, evaluating the current model's performance across diverse tasks and data would be beneficial. An area of interest is how well the performance of the proposed Transformers scales with significantly longer input sequences. Drawing inspiration from successful weight-sharing methods in NLP, as seen in prior work (*e.g.* Kitaev et al. (2020)), might also enhance the accuracy of our models on NLP tasks. We anticipate that our approach will prove particularly valuable for Cache Augmented Generation (CAG) architectures, including recent advances such as that proposed by Chan et al. (2025). A key benefit lies in its ability to substantially decrease the memory footprint associated with the KV cache.

**Discussion of Limitations.** While the improvements in parameter and computational efficiency are modest, we observed significant reductions in training and inference time on tasks like image classification with Tiny ImageNet. We believe that the slight accuracy decrease could potentially be mitigated by extending the training duration.

**Broader Impacts.** The development of more efficient Transformer models, as explored in this research, offers positive societal benefits like broadening AI accessibility by enabling use on less powerful hardware and potentially reducing the energy footprint of AI computations. The growing ease of deploying more powerful AI is a welcome development. However, it's crucial to carefully examine how these advancements are put into practice. Our findings align with existing research in this area and do not introduce any new negative implications beyond what is already documented about these types of models.

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

# A  APPENDIX

Figure 7 shows the loss over time for the synthetics tasks. Figure 8 displays sample attention maps. It should be noted that the attention maps of the KV transformer exhibit symmetry around the line $y = x$. Notable patterns can be observed within the attention maps. For instance, in the reversing task, the QKV model has learned to take care of the token located at the flipped index of itself. However, it also allocates some attention to values near the flipped index. This behavior arises because the model does not require precise, strict attention to solve this problem, but rather benefits from an approximate, noisy attention map. Figure 9 shows the code to compute and normalize the self attention map, plus visualization of maps.

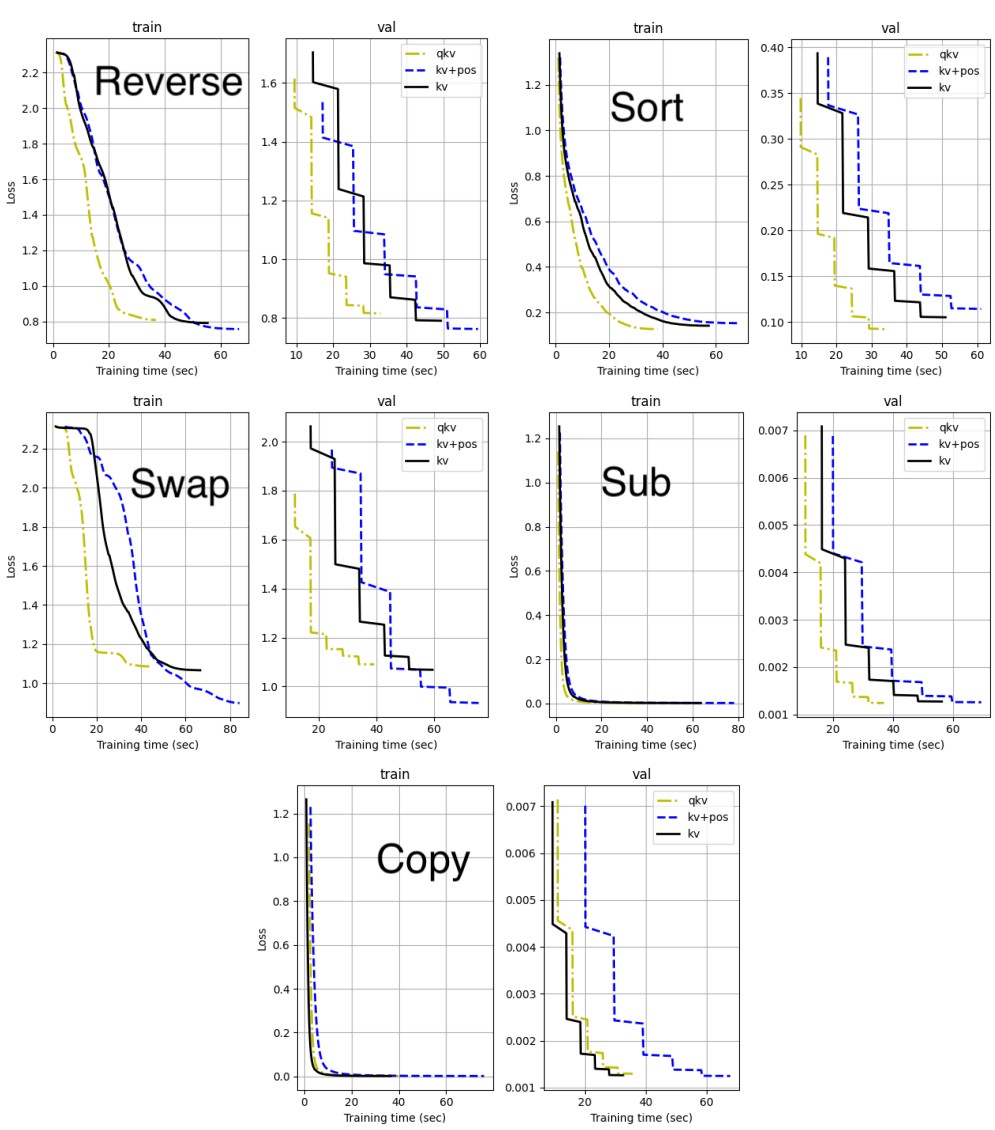

Figure 7: Loss over time for the synthetics tasks.

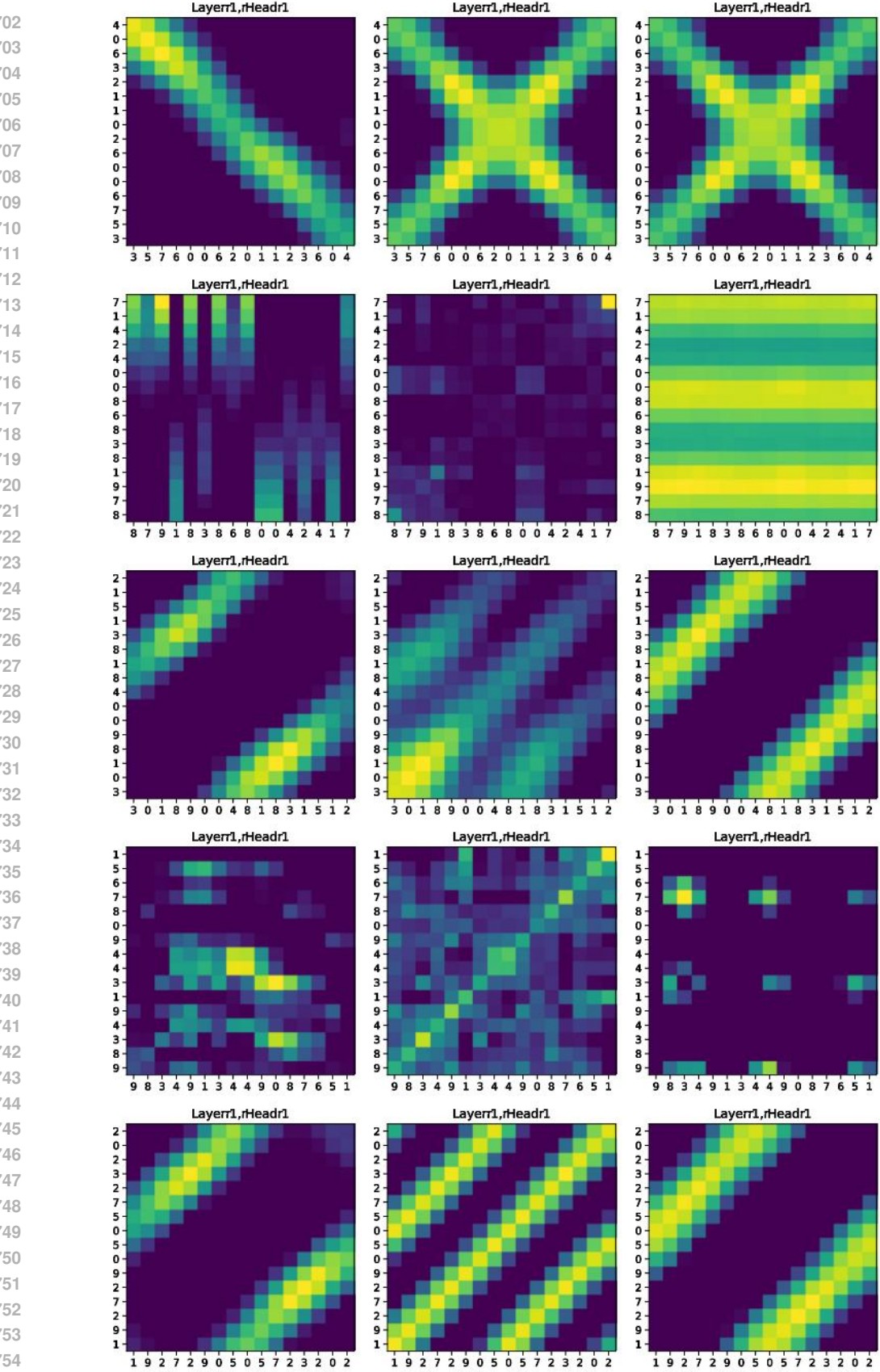

Figure 8: Attention maps over synthetic tasks. Rows from top to bottom: Reverse, Sort, Swap, Sub, and Copy. Columns from left to right: QKV, KV, and KV+Pos.

```python
def normalize_kkt_diagonal(K):
    """Normalizes the diagonal of K @ K^T."""
    S = K @ K.T  # Calculate K @ K^T
    diagonal = torch.diag(S)
    scaling_factor = torch.mean(diagonal)  # Or torch.linalg.norm(diagonal) or other scaling factor.
    normalized_S = S.clone()

    # Iterate and fill each diagonal element
    for i in range(normalized_S.shape[0]):
        normalized_S[i, i] = diagonal[i] / scaling_factor

    return normalized_S, S

# Example Usage:
n = 10
h = 30
K = torch.randn(n, h)  # Generate a random K matrix
S_normalized, S = normalize_kkt_diagonal(K)
print("Normalized S:\n", S_normalized)
```

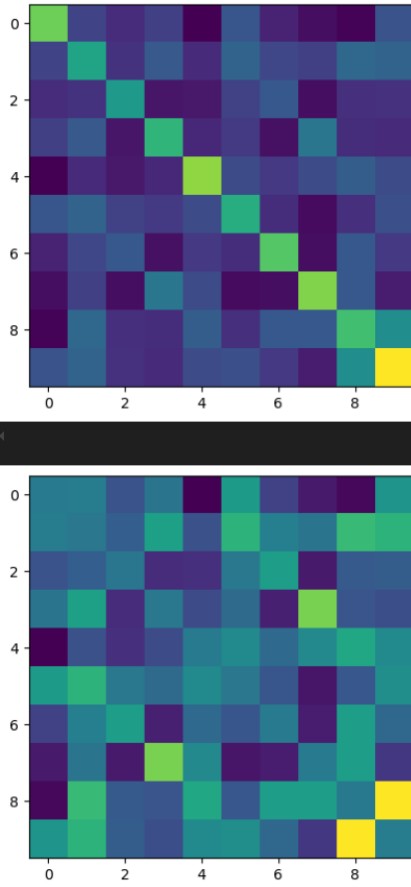

Figure 9: Top) Code to compute and normalize the self attention map. Bottom) un-normalized and normalized (bottom) attention maps.