# OpenReview forum: "From QKV to K/KV: Investigating Minimalist Attention Mechanisms"
_ICLR.cc/2026/Conference — ICLR 2026 Conference Desk Rejected Submission_

### Official Review · Reviewer_8Qke · 2025-10-25

**Soundness:** 2
**Presentation:** 2
**Contribution:** 2
**Rating:** 4
**Confidence:** 3

**Summary:**

This paper investigates whether all three components of the standard Query-Key-Value (QKV) attention mechanism are necessary in Transformers. The authors propose two simplified variants: KV Transformer (which removes the query projection and uses symmetric attention) and K Transformer (using a single projection). Additionally, they explore incorporating 2D positional encoding to introduce asymmetry. Experiments across 14 tasks spanning synthetic problems, vision (image classification and anomaly detection), and NLP (number/character generation and translation) reveal task-dependent results: modified Transformers perform comparably to or better than standard QKV on vision tasks while underperforming on NLP tasks.

**Strengths:**

1. Clear motivation and systematic exploration: The paper addresses a fundamental but under-explored question about the necessity of distinct Q, K, and V projections. The systematic evaluation across diverse task types (synthetic, vision, NLP) with multiple datasets provides broad empirical evidence for the task-dependent nature of attention mechanisms.
2. Practical efficiency gains with acceptable trade-offs: The proposed variants reduce parameters (KV: 2/3, K: 1/3 of QKV) and computational cost, with demonstrated wall-clock time improvements (e.g., 40→32 minutes per epoch on Tiny ImageNet). The authors thoughtfully discuss the trade-off between efficiency and performance, recognizing that applications with different constraints can benefit from different variants.

**Weaknesses:**

1. Weak theoretical analysis and understanding of failure modes: The paper provides minimal insight into why KV/K attention fails on NLP tasks while succeeding on vision. The explanation that "token order influences meaning" is superficial. The symmetric attention matrices may lose important directional information needed for sequential tasks, but this is not rigorously analyzed. The observation about diagonal activations dominating (Section 3.2) mentions failed normalization attempts without deeper investigation. Understanding fundamental limitations would strengthen the contribution.
2. No evaluation on large-scale pre-trained models (e.g., BERT-scale or larger), which would better demonstrate practical relevance.
3. Short training durations (2-50 epochs) may not reveal convergence behavior or final performance gaps.
4. Missing comparisons with MQA/GQA—the closest existing methods.
5. The Tiny ImageNet experiment shows 22.9% vs 23.6% accuracy, a non-negligible gap. The claim that "continued training could close the gap" lacks supporting evidence. These limitations constrain the impact and generalizability of the findings.

**Questions:**

See weakness.

---

### Official Review · Reviewer_kJ99 · 2025-10-28

**Soundness:** 2
**Presentation:** 3
**Contribution:** 2
**Rating:** 2
**Confidence:** 3

**Summary:**

The paper removes Q or ties Q and K to form symmetric attention variants (K, KV, plus K+Pos and KV+Pos that add 2D positional components) and tests them on small benchmarks. It claims similar or sometimes better results in vision and mostly worse results in NLP, with modest parameter and projection-layer compute savings.

**Strengths:**

The idea is simple. Setting Q=K and using A = softmax(α K Kᵀ) is easy to implement. The attempt to inject 2D positional signals to break strict symmetry is straightforward.

The computation and parameter counts for projection layers are clearly tabulated. The K variant uses roughly one third of the projection parameters relative to standard QKV.

On Tiny ImageNet there are cases where K outperforms QKV. Training time is also shorter for K in the reported setup.

**Weaknesses:**

1. Sharing or tying Q and K has appeared in prior discussions. The paper lists related ideas such as weight sharing and MQA or GQA but does not offer direct head-to-head comparisons. The contribution feels incremental without a compelling empirical advantage over close baselines.

2. The main table focuses on projection-layer complexity and sidesteps the quadratic attention costs. KV+Pos introduces an extra n²m path and can become more expensive at practical sequence lengths. There is no full FLOP breakdown, memory profile, or latency study under realistic settings.

3. With K Kᵀ the diagonal terms tend to dominate. The paper acknowledges that its normalization attempts do not bring consistent gains. This signals an expressivity limitation of the core proposal.

4. Most benchmarks are tiny. Training is from scratch with short schedules on a single 1080 Ti. Modern practice relies on large pretraining and transfer. NLP experiments use very small datasets such as Multi30k and tiny Shakespeare, which weakens any generalization claim.

5. Capacity differs across variants since parameter counts change. A fair test should fix either parameters or FLOPs. The paper reports averages across few runs but lacks standard deviations, significance tests, or capacity-matched studies.

6. The average column in a main table excludes Tiny ImageNet, yet the text highlights Tiny ImageNet wins. This creates a cherry-picking impression.

7. The paper notes that K and KV often underperform on generation tasks and that K or K+Pos perform so poorly that they are dropped from later experiments. There is no convincing root-cause analysis or remedy for language tasks.

8. The write-up suggests sinusoidal 2D positions plus a learned projection from an n×n×m tensor, but another part says the 2D positional embedding is not learned. Readers cannot tell which parts are trainable. Clearer ablations are needed.

9. The paper hints at applicability to tasks such as semantic segmentation and to scenarios like KV-cache efficiency or CAG. It does not provide experiments to support these claims.

10. Many hyperparameters and training details are underspecified. Runs are few. There is no code or checkpoints. The setup invites variance concerns that the paper does not address.

11. Only two epochs and very simple rules. This is not a stress test for attention structure. Conclusions from these numbers are limited.

12. On MNIST and CIFAR the gaps are small. Tiny ImageNet shows wins for K, but only two runs are reported. The text speculates that longer training might change gaps without evidence.

13. The method relies on ResNet34 features then does simple set classification. This setup dilutes the effect of the attention variant. It is only weakly connected to the paper’s central claim.

14. KV converges slower on numbers and tiny Shakespeare. KV+Pos hurts translation. The paper does not probe why symmetry or read-write coupling harms language modeling.

15. The paper cites weight sharing, MQA, GQA, and DHA-style ideas but omits direct comparisons. At minimum, a controlled GQA or MQA comparison with matched compute and capacity is necessary, along with modern linear-attention baselines.

**Questions:**

For long sequences, when does the n²m path of KV+Pos pay off in end-to-end speed or accuracy. Show measured curves, not only asymptotic conditions.

Why are K and KV weak on language tasks. Is the issue strict symmetry, the loss of separate query semantics, or interference between reading and writing through shared keys. Provide diagnostics or theory.

Why is Tiny ImageNet excluded from the reported averages. Please provide averages that include and exclude it.

Will you compare against GQA or MQA directly with equalized parameters or FLOPs.

Which parts of the positional pipeline are learnable. Provide a clear diagram and ablations isolating each component.

Add capacity- or FLOP-matched comparisons with variance bars and significance tests.

Include direct baselines such as GQA, MQA, and a strong linear-attention method.

Measure full compute, memory, and latency for realistic lengths, and show scaling curves.

Analyze diagonal dominance and symmetry in theory, then propose a principled fix such as gated off-diagonal boosting or explicit anti-diagonal regularization.

Release code, configs, and checkpoints.

---

### Official Review · Reviewer_Z97p · 2025-10-30

**Soundness:** 1
**Presentation:** 3
**Contribution:** 2
**Rating:** 2
**Confidence:** 5

**Summary:**

This paper investigates whether separate query/key/value weights are necessary in the standard self-attention mechanism. It proposes two variants: the KV transformer that merges the query and key weights, and the K transformer that uses only one set of key weights for all three of queries, keys, and values. On a variety of vision and language tasks (especially synthetics), they find the variants are similarly effective to default self-attention. However, the variants underperform on _natural_ language tasks. Their takeaway is that not all tasks require separate QKV weights.

**Strengths:**

- Reasonable well-scoped scientific question, and it was a good idea to add position encodings to break symmetry
- Broad empirical investigation on a variety of tasks, from vision to language and synthetics
- High clarity, experiments and ablations are well-described and the paper is well-written

**Weaknesses:**

- Undertrained baselines: 66% on CIFAR-10, for example, is too low to make meaningful comparisons across models. It's notoriously hard to train ViTs on small datasets without other interventions, like extensive data augmentation or initialization (eg, mimetic initialization).
- Insufficient scale: it's very unlikely that the results will generalize from such simple tasks -- it would be much more convincing to see something at the ImageNet or GPT-2 scale. While this is a nice workshop paper, I don't think it's appropriate for presentation at ICLR.
- Limited analysis of gap between NLP and vision tasks for this intervention (though the attention maps in the appendix are a nice touch)

**Questions:**

Do you have any ideas on the gap between vision and NLP tasks for the KV/K-transformer variants? Any examples of things that QKV attention can provably do that KV- or K-attention cannot?

---

### Official Review · Reviewer_kPQF · 2025-10-30

**Soundness:** 3
**Presentation:** 3
**Contribution:** 2
**Rating:** 4
**Confidence:** 4

**Summary:**

This paper explores whether the standard query–key–value (QKV) decomposition in Transformers is truly necessary for strong performance across domains. The authors introduce two simplified variants:
1. KV Transformer, which uses only keys and values (Q=K), resulting in symmetric self-attention.
2. K Transformer, which uses a single projection for all roles (Q=K=V).

They also introduce a variant that restores asymmetry by adding 2D positional encodings (KV+Pos, K+Pos) directly into the attention matrix. Across a suite of synthetic, vision, and NLP tasks, they find that:
- KV/K perform comparably to QKV in many vision and synthetic tasks. Note that TinyImageNet results appear to be mislabeled, which I will describe further below.
- NLP tasks show a significant performance drop without the full QKV decomposition.

Their modified models show reduced parameter counts, and the K and KV show increased computational efficiency, as do the K+Pos and KV+Pos for short sequence lengths (at longer sequence lengths, the additional positional quadratic term outweighs the gains in projection efficiency).

The results suggest that for some modalities, particularly static image processing tasks, QKV may not be necessary, and a more symmetric and parameter efficient variant can be suitable.

**Strengths:**

- The approach is simple and well-motivated. I was surprised that it had not already been examined in past literature. The authors situate their contribution well within the field in the Related Work section.
- Multiple domains and tasks are tested, showing both strengths and weaknesses of the approach, providing a balanced and informative set of results for the reader
- The approach could help increase efficiency in vision applications, or other static and symmetric tasks

Minor:
- Figure 1 is very nice

**Weaknesses:**

1. The computational complexity analysis is somewhat flawed because it ignores the quadratic attention operation. The authors mention this, but it does hide the relative improvements in efficiency.
2. The experiments are toy-level. The author appears to be working with a limited compute budget, so I do not strongly fault them for this. However, it does limit the relevance of the results to the broader community. Other authors would have to reproduce their results on larger datasets before strongly believing them.
3. Benefits are largely restricted to the synthetic tasks and vision tasks, with deficits seen for NLP. This is fine, but for the vision task, I would like to therefore see a larger scale experiment. Even Tiny-ImageNet is small for ViTs, which lack inductive bias, and therefore strongly overfit (more so than CNNs). Since the authors show promising results here, it would be very informative to know how the models scale to the full ImageNet dataset. Results are also missing for K+pos and KV+pos, which is disappointing since these models are most applicable in the vision setting where sequence lengths are shorter and therefore efficiency gains are still expected.
4. The TinyImageNet results are misreported in either the results or discussion, and I am not sure which. Table 3 and Figure 2 show best performance for K, followed by KV, and QKV. I find this hard to believe, since this performance is also associated with better fits to the training data. It is nearly inconceivable that the K model would yield lower training loss than the QKV model. The discussion also says that these models show modest decrements that could be improved with further training.
5. Given the 4), it appears that the architecture does not show any improvements for any reasonably large-scale task.

Minor:
6. I do not particularly like the anomaly detection task because the easiest thing to do is just compute similarity of the representations and then take the one whose average similarity is minimized. i don't see what the transformer has to do with this when there is a pre-trained image encoder that does all the heavy lifting.
7. Some of the figures are very low-res.
8. The figures lack consistency. QKV, KV, and K should each use a single color across figures, and +Pos should introduce a variable linestyle or marker.
9. Figure 5 should be consistent with others and use epochs as the x-axis. The choice to use training time here only is completely out of the blue and not motivated in the text.

**Questions:**

- Can you add the shared quadratic term in the computational complexity figure, and also add empirical FLOPS for multiple sequence lengths? This can illustrate how the benefits cache out in actual architectures, dependent on sequence length, in terms of % reductions in FLOPS.
- Can you address weaknesses 3) and 4) to help me understand your results better?
- Please fix the broken link to the TinyImageNet dataset

**Details Of Ethics Concerns:**

No discussion of LLM usage, as mandated by ICLR policies

---

### Official Review · Reviewer_K6Mw · 2025-11-01

**Soundness:** 2
**Presentation:** 2
**Contribution:** 1
**Rating:** 2
**Confidence:** 5

**Summary:**

This paper investigates the necessity of the standard Query, Key, and Value (QKV) projections in the Transformer self-attention mechanism. The authors propose and evaluate two "minimalist" variants:

1. KV Transformer: This model removes the Query (Q) projection and ties the query and key representations, using the formula $A=Softmax(\alpha KK^{T})V$.

2. K Transformer: This model goes further, using only a single projection (K) for query, key, and value: $A=Softmax(\alpha KK^{T})K$.

The authors test these variants across a range of synthetic, vision, and NLP tasks. The core findings are that these models offer reduced parameter counts and computational complexity and perform on par with the standard QKV Transformer on vision tasks but underperform on NLP tasks.

**Strengths:**

1. Fundamental Question: The paper questions whether the full QKV formulation is universally required, challenging a default assumption in Transformer design.
2. The minimalist models (KV and K) were competitive or even superior in vision tasks, while the full QKV model was clearly better for NLP.

**Weaknesses:**

1. Modest Overall Gains: As the authors rightly concede, the overall improvements in parameter and computational efficiency are "modest". This is because the self-attention projections, which are the paper's sole focus, "only represent about a third of a typical Transformer's total parameter count". This limits the practical impact of this specific optimization.

2. Comparison to Other KV Cache Methods: The paper mentions that its approach is "orthogonal" to existing methods like multi-query and grouped-query attention (GQA). However, GQA is a widely-adopted and highly effective standard for reducing the KV cache with minimal quality loss. The review would be much stronger if it experimentally compared the proposed KV/K models against GQA. Without this, it's unclear if this new approach is a useful complement or simply a less effective alternative.

3. Overlooks Conventional Optimizations: The paper frames its investigation as a path to efficiency but jumps directly to the radical step of removing entire Q or V projections. It fails to discuss or compare its approach to more common and established optimization techniques, such as reducing the number of attention heads or decreasing the head dimension. The paper does not justify why removing entire projections is a more promising research direction than simply tuning these other well-known hyperparameters.

4. Theoretical vs. Practical Complexity: The complexity analysis in Table 1 is based on the conceptual, mathematical formulation of three separate projection matrices ($W_q, W_k, W_v$). In practice, many modern Transformer implementations fuse these into a single, efficient $W_{QKV}$ matrix multiplication. The paper does not discuss this common optimization, which somewhat muddles the real-world-speedup argument. The reported wall-clock time gains are valid but may not be solely due to the parameter reduction vs. other implementation-level factors.

5. Limited Applicability: The clear performance drop on NLP tasks indicates that these minimalist models are not a general-purpose replacement for the standard Transformer. Their utility seems to be niche, primarily for non-sequential tasks where attention symmetry is not a hindrance.

**Questions:**

See weaknesses.

---

### Note · Program_Chairs · 2026-01-17
**Submission Desk Rejected by Program Chairs**

The following references in this submission do not refer to real documents and/or have major errors in bibliographic information:

 Shen He and Yankai Liu. Model compression techniques for transformer models: A survey. http: //www.surveyx.cn/assets/img/pdfs/Computation\ and\ Language/model.pdf, 20